

# How to explain variations in sea cliff erosion rates? Insights from a literature synthesis

Prémaillon Mélody[1], Regard Vincent[1], Dewez Thomas J.B.[2], and Auda Yves[1]

[1]GET, Université de Toulouse, UPS (OMP), CNRS, IRD, 14 avenue Edouard Belin, 31400 Toulouse, France
[2]BRGM, F-45060 Orléans, France

**Correspondence:** Premaillon Melody (melody.premaillon@get.omp.eu)

**Abstract.**

Rocky coast erosion (i.e. cliff retreat) is caused by a complex interaction of various forcings that could be marine, subaerial or due to rock mass property. It turns into variable erosion rates (over 4 orders of magnitude at least, from 1 $\mathrm{mm.yr^{-1}}$ to 10 $\mathrm{m.yr^{-1}}$). While numerous local studies exist and explain erosion processes on specific sites, there is a lack at global scale. In order to quantify and rank the various parameters influencing erosion rates, we compiled existing local studies in a global database called GlobR2C2 (for Global Recession Rates of Coastal Cliffs). This database records erosion rates, cliff setting and measurement specifications; it is filled from peer reviewed articles and national databases. In order to be homogeneous, marine and climatic forcings were recorded from global models and reanalysis. Up to now, GlobR2C2 contains 58 publications which represents 1530 cliffs studied and more than 1680 erosion rate estimates. A statistical analysis was conducted on this database to explore links between erosion rate and forcings at global scale. Rock resistance, inferred through Hoek and Brown (1997) criterion, is the strongest signal explaining variation in erosion rate. Median erosion rates are of 2.9 $\mathrm{cm.yr^{-1}}$ for hard rocks, 10 $\mathrm{cm.yr^{-1}}$ for medium rocks and 23$\mathrm{cm.yr^{-1}}$ for weak rocks. Concerning climate, only the number of frost days (number of day per year below 0°C) for weak rocks shows a significant, positive, trend with erosion rate. Every other relations with both climatic and marine forcings are very spread and non-significant.

*Copyright statement.* TEXT

## 1 Introduction

Rock coasts are characterized by dynamically linked cliff retreat and shore platform erosion. By comparison between continental and coastal cliffs, it is clear that the presence of the sea is a fundamental driver (Fig.1). But, as Moses and Robinson (2011) felt, our understanding of their dynamics and our ability to predict their evolution over time remains severely limited".Kennedy et al. (2014) emphasize the growing number of quantitative studies, allowed by development of new investigation methods like lidar techniques. According to their analysis, a reassessment of cliff retreat rates is needed. Hence, the purpose of this paper is to take advantage of this growing corpus of data in order to quantitatively analyse cliff erosion drivers.



These drivers can be divided in three groups, depending on their nature (Fig.2). The first group concerns marine forcings. Waves attack and weaken cliff base, sometimes carving a notch, leading to cliff instability and subsequent collapse (e.g., Benumof et al., 2000; Caplain et al., 2011). It is a common assumption in coastal landscape evolution model and led to the debated term 'wave cut platform' (Anderson et al., 1999). Debris aprons are then removed by sea action, allowing for renewed

wave attack at cliff base. Wave assailing force depends on wave energy dissipation over the shore platform (e.g., Sunamura, 1992; Trenhaile, 2000). The wider and shallower the platform is, the lower is the remaining wave power at cliff foot. Hence platforms can be regarded as natural defences against wave attack to the cliff. The shore platform is evolving under marine forcing like, wave agitation and associated shear stress (e.g., Sallenger Jr et al., 2002; Stephenson and Kirk, 2000; Sunamura, 1992; Trenhaile, 2008, 2009), or tide-induced wetting and drying cycles (Kanyaya and Trenhaile, 2005; Stephenson and Kirk,

2000). The second group of drivers is rock mass properties that are supposed to have a strong influence on cliff evolution (Mortimore and Duperret, 2004). The rock mass behaviour depends on its lithology, structure, fracturing and weathering (e.g., Cruslock et al., 2010). The third group of drivers is made of subaerial processes. Climate through precipitations, temperature or frost occurrences (e.g., Dewez et al., 2015) may either provoke cliff instability or prepare it by physical and chemical weathering (Duperret et al., 2005).

All these have been proven to be efficient in their own way in cliff retreat phenomena, but their relative importance is perceived differently in the studies (Fig.2), likely because of the small spatial extent or the authors' field of expertise. Some attempts exist at local scale to rank the different drivers (e.g., Earlie et al., 2015; Lim et al., 2010) but they can hardly be upscaled.

Some studies aimed at quantifying cliff retreat rates at the regional scale i.e. coastal sections of several tens to hundreds

of kilometres (e.g., Gibb, 1978; Abellán et al., 2009; Hapke et al., 2009; Moses and Robinson, 2011; Perherin et al., 2012). In terms of retreat drivers, they are inconclusive because: (i) they do not analyse the contribution of each driver; and (ii) they characterise a narrow range of forcings (e.g. climate, homogeneous lithology . . . )

In order to pass through bias inherent to individual approach, studies at global scale have been conducted. They are often based on qualitative morphometry. For example, the classic study of Emery and Kuhn (1982) interprets cliff profile morphology

as a function of cliff top and toe composition and marine and subaerial relative process efficiency. The only global, quantitative, data set was produced bySunamura (1992), on the basis of quantitative studies published prior to that date. Sunamura's database was only used by Woodroffe (2002) to evaluate ranges of erosion rates for different lithologic type. Up to now, those rates have never been related to environmental factors.

Since Sunamura (1992)'s compilation, 26 years ago, many new quantitative studies have been published (Fig.5). They took

advantage of several technological changes in that time interval. Airborne and terrestrial lidar as well as structure-from-motion (sfm) have revolutionized ad hoc surveys in geosciences, making precise geometric information available where and when required. National mapping agencies released their aerial photography archives online. These provide contemporary surveys with a historical context. Software developments afforded massive 3D processing capabilities, even to non-specialists. So quantitative site studies are now addressing cliff face erosion style at centimetre-scale (e.g., Dewez et al., 2013; Earlie et al.,

2015; Gulayev and Buckeridge, 2004; Letortu et al., 2015; Rosser et al., 2007; Young and Ashford, 2006). This high spatcial



accuracy is nowadays added to high time resolution up to 20 minutes Rosser (2016). Cliff recession phenomena have never been so well defined in space and time. It is now time to sort through possible processes generating cliff responses.

We updated Sunamura (1992)'s data set into the new database GlobR2C2, for Global Recession Rates of Coastal Cliffs, by taking the benefit of all the existing case and regional studies and built a worldwide cliff recession database. This database is used in a new approach to link erosion rate and external forcings and study their relative efficiency. The benefits of this global approach is to erase local specifics and to look at global trends. The links between cliff retreat and environmental parameters were explored statistically. The synthetic database approach however has the limit that it compiles the information available for all studies at once. In that sense, it reduces information to the largest common denominator. The main goals of this paper are to: (i) make a review of online literature in English, French or Spanish language from peer-reviewed or national database providing cliff retreat rates; (ii) link a dependent variable: erosion rate to independent variables: cliff and meteo-marine settings. The analysis demonstrates the predominance of factors leading to cliff retreat.

## 2 Method

### 2.1 Study design

The main goal of this study is to link erosion rate to external forcings at global scale. Those data exist in peer reviewed journal articles and national databases. Peer reviewed articles were chosen to be the source of cliff description and erosion rate value and settings. However, boundary conditions are often reported in a very heterogeneous fashion. Those information can either be completely lacking, incomplete or can be described in very heterogeneous ways. To overcome this issue, external global database were used to homogenise forcings. They provides homogeneous and reputable information for cliff height, sea condition and atmospheric climate.

The different steps of the study that are going to be described in the next paragraphs are: (i) creation and filling of a database with raw data, (ii) post-processing on database fields in order to tidy data (iii) statistical exploration of links between erosion and forcings.

### 2.2 Database design

To organise the disparate knowledge reported in the literature, a rigorous analysis framework is an absolute necessity upstream of any data capture. We opted for a relational data base framework whose architecture was conceived according to the Merise method (Tardieu et al., 1985). Merise provides a formal methodology to describe entity-relationship data models. It helps conceptualize data groups (entities), their properties (fields), specifying the existence of relationships between tables, their direction (one-way, two-way, origin and destination) and their number (one entry relates to one or to multiple entities). This conceptual exercise allows to minimize data capture and data redundancy, to flag possible information replicates and limits ill-conceived relationships. The database structure was implemented in OpenOffice Base that can be processed in R via SQL



queries. Only the geographic fields (cliff location) were digitized in GoogleEarth and exported into shapefile with a key code or primary key linked to the relational database (in the sense of data science analysis).

Here, data was structured with two objectives in mind: (i) compiling original information and faithfully tracing publications source, and (ii) anticipating analytic queries of the database designed to answer geomorphological questions. The database is

structured to keep track of information relative to publications, sites, measurements and contextual information of the cliffs, or their environment. Specific care was taken to separate original data from information derived by us, and distinguish between article information from auxiliary data sets. The database contains entities coming from tree type of sources: raw data from papers, raw data from gridded data and tidy covariates.

The final conceptual data model contains 11 entities and 76 attributes. A conceptual model is given in (Fig.3). Entities refer

to publications (*Publication and Author*), cliffs ( *Cliff, Lithology, Geotechnical parameters, cliff height*); erosion rate measure (*Measure*) and to forcing (*Climate, Swell, Tide*). Information contained in each entity come from publication except entities concerning forcings and Geotechnical parameters which come from external sources. The relation between the different entities are given by the action verbs and the numbers represents the cardinality of the relation (eg. 1 cliff can correspond to 1 or N erosion rate measure, cardinality 1,N).

## 2.3   Database information fields

### 2.3.1   Raw data extraction: From publication and national databases

GlobR2C2 (Global Recession Rates of Coastal Cliffs) database v1.0 was populated with data coming from two main type of published sources: published peer-reviewed English journal articles, and official but non-peer-reviewed studies arising from official organizations (e.g. CEREMA French risk survey) in English French or Spanish language. Journal articles were selected

when they proposed quantified value of cliff recession rates and described the quantification method. The search was initiated with bibliographic web search engines (Web of Science, Google Scholar) and expanded using citations therein. We admit that some references may have escaped our attention. We are keen to expand the database further with the contribution of the community. The version presented in this article is version 1.0. compiling references up to 2016.

### 2.3.2   Cliff and lithology description

Cliff and lithology entities contains information relating to cliff settings: height, length, lithology, fracturing, weathering, folding, bedding etc.

Describing the rocks of a cliff sector is a bit of a tricky issue. They may vary from place to place laterally and vertically. They may also be fractured or weathered differently and authors do often not apply a very strict formalism to report them. In front of the broad diversity instances, we synthesized information in the following manner. A lithological name fills the

"lithology" entity and a position field records rock position along the cliff (head, toe or overall). Additional descriptions were copy/pasted in comment fields in order to preserve a trace of the original description. By comparison rock state (weathering, folding, faulting, bedding etc.), is rarely mentioned. This could be because the cliffs do not present any such characteristics,





or because authors did not think relevant to mention it. Moreover, parameters describing rock state are either complex or technically expensive to describe and quantify or outside the authors' scientific field of expertise. They were characterise with a Boolean value to be integrated in the database. True refers to the presence of fracturing/weathering mentioned in the paper, false otherwise.

### 2.3.3  Cliff location

Cliff location is entered as a geographic information. Studied cliff site extent was digitized from to publication information and mapped using Google Earth . A primary key links this geographic file to the database.

### 2.3.4  Measure description

The measure entity contains the erosion rate values and measurement methodology (how erosion was measured, for how long etc.). Erosion is most of the time provided as an erosion rates in meters per year, occasionally as finite retreat (in meters), minimum and maximum erosion rate or eroded volume (in cubic meters).

Cliff retreat measurement errors and time spans were also recorded. Indeed, measuring sea cliff erosion presents a wide range of techniques. Those techniques vary largely in terms of: (i) accuracy, from field observation and "expert" estimates (e.g., May, 1980) of volume loss to lidar (e.g., Dewez et al., 2013) ; (ii) time period surveyed, from twenty minutes (Rosser, 2016) to thousands years (Choi et al., 2012; Regard et al., 2013); and (iii) Spatial extent, from tens of meters (e.g., Letortu et al., 2015) to kilometres (Gibb, 1978; Hapke et al., 2009) Moreover, these measurements can be divided into three classes of methods: 1D, 2D or 3D.

1D cliff retreat measurement techniques correspond to retreats calculated on single transects. Typically, they correspond to measure done with peg transects recording the cliff toe retreat or transects on aerial photographs to quantify cliff-top retreat (Kostrzewski et al., 2015; Lee, 2008; Pye and Blott, 2015). 2D measurements are mostly based on aerial photograph comparison. They either quantify the area lost between two aerial photographs campaigns or average numerous transects (Costa et al., 2004; Letortu, 2013; Marques, 2006). 3D techniques record the evolution of the cliff face and quantify volumes (e.g., Letortu et al., 2015; Lim et al., 2005; Rosser et al., 2007). The oldest method is rockfall inventory (rockfall volume estimation based upon size of debris or scar, (e.g., May, 1971; Orviku et al., 2013; Teixeira, 2006) but now the two most used methods are lidar and SfM.

### 2.3.5  CEREMA French national dataset: a particular case

The French CEREMA institute published a systematic national coastal cliff recession inventory (Perherin et al., 2012) based on aerial photograph comparison every 200 meters stretch of cliff along the entire French metropolitan coastline (1800 km of rocky coast). This rich systematic dataset was obviously included in GlobR2C2 but raise two caveats. On the one hand, the CEREMA study introduces a strong spatial bias for French oceanographic and climatic conditions in the database observation records. This situation may risk to polarize analytical results but was recognized beforehand and specifically treated to prevent such





bias. On the other hand, being a systematic study for every stretch of coastal cliff around the country, it makes it more robust to scientific and funding biases. Research funds are often sought for areas combining coastal threats with societal interest. Coasts with higher recession rates are therefore more often sampled, while quiet stretches of coastlines remain in the shadow. Including this data therefore provides a more representative set of values existing along coastlines. Among little studied sectors

it represents this CEREMA study contains hard rock coastal stretches (e.g. hard Proterozoic granites from French Brittany) and erosion rates lower than the study's detection threshold.

Based on historical aerial photograph archives, CEREMA acknowledges that photographs quality limits the detectable cliff recession to rates higher than 10 cm/yr. Below this value, they deem recession rates as undetermined. We chose to record those undetermined values in the database but not to use them in the statistical analysis . We discuss this choice in the following.

**2.3.6   Tide**

The tidal range describes the water column height at cliff foot and produce wetting and drying cycles (Kanyaya and Trenhaile, 2005). Rather than referring to difficult use tidal records from tide gages, tidal modelling was performed with FES 2012 software (Carrère et al., 2012). This model which gives all the constituents of the harmonic tide analysis. 8 harmonics were considered: M2, N2, K2, S2, P1, K1, O1. The model produces time series of sea water height within a regular grid of 0.25

degree between two dates. The tide is computed for each study location for two entire years, of which is extracted the mean amplitude over N cycles (i.e. height difference between successive high and low tides).

**2.3.7   Waves**

Wave properties were extracted from ERA-interim reanalysis dataset (Dee et al., 2011). This gridded data has a pixel size of 0.75 degree. Temporally, data spacing is 6 hours during the 1979-2016 period. Wave assault was characterised both in terms

of mean agitation and extreme events. Three mean parameters characterise wave assailing force: significant wave height of combined swell and wind, wave period and wave direction. For swell characteristics, mean significant wave height and wave period characterise the average sea agitation. The wave direction value records the most frequent wave direction for the duration of the reanalysis period (1979-2016).

Anticipating that mean sea state values may be deceptive metrics, a record of extreme events was also described. Those

events were characterised by the 95 % percentile of wave significant height. To complete this quantile value, the number of storms experienced at each cliff site was calculated between 1979 and 2016 according to (Castelle et al., 2015).

**2.3.8   Climate**

Climatic information was extracted from Climate Research Unit data between 1961 and 1990 Mitchell and Jones (2005). The grid size is 0.5 degree, at monthly time step. Chosen parameters likely to influence erosion rate are mean annual rainfalls, mean

monthly temperatures and number of freezing days (number of days per year bellow 0°C).



### 2.3.9 Cliff height

Cliff height appeared to be often missing. Filling this value is not straightforward because cliff height can be strongly variable along the surveyed cliff. Nevertheless, in order to provide a robust estimate, a mean cliff height was extracted from the 8" global DEM (GMTED2010, Danielson and Gesch, 2011). Cliff height extraction consisted in computing a buffer around the

cliff extension shapefile, in which the mean value of non-zero pixels (corresponding to the sea) is computed. To assess the accuracy of these cliff height estimates, they were compared against those rare values presented in publications. Computation is close to value given in publication with a root mean square error of 19 m at global scale. It is quite good for a first attempt at the global scale, probably not so far from the cliff height accuracy in the publications.

### 2.4 Tidying the covariates: from database fields to predictors

The first purpose of the database is to collate raw data from original sources in the most traceable manner. This data does not necessarily report information in an easily accessible fashion. This may be because: (i) fields translate different realities (e.g. recession rates vs retreat values or recession rates relate to profile-specific recession rate or to kilometre long cliff sections), (ii) value instances of a field is too broad and needs summarizing in fewer categories (e.g. lithology). Thus, post processing was applied to the database in order to make it more homogeneous and more readily usable for statistical analysis.

### 2.4.1 Integration of punctual records

We mentioned earlier that measurement techniques were either 1D, 2D or 3D. These methods do not reflect exactly the same processes and a choice was made to force all measurements to report 2D type measurements homogeneously. This 3D measures in $m^3.yr^{-1}$ were divided by cliff face surface in a cliff top equivalent retreat in $m.yr^{-1}$. 1D measurements do not average information spatially. Cliff retreat is a stochastic process in time and space and 1D measurements profiles may happen to

quantify erosion on a particular high or low erosion transect. Erosion rates of the transect measures were therefore averaged for a unique study, cliff and period of time in order to limit the risk of over/under-representation.

### 2.4.2 Field unit conversion

Original data may be provided in different ways (for example the time span between 2 measurements may be given by a duration or start and end dates). As often as possible this information is summarized in a single duration field with homogeneous unit.

This lists the operations performed:

- To obtain a duration in years, the fields measure duration [year], measure beginning and measure ending [date] were merged together

- Retreat [m] and eroded volume [m3] were normalised to retreat rate [$m.yr^{-1}$].

- The mean cliff height is either obtained from a cliff height mean field or as the mean between height min, height max

[m].



– The error (m/yr) is a compilation of error value and error type.

### 2.4.3  Average site climate

Some explanatory variables were strongly correlated with each other (e.g. wave period vs wave significant height). This redundant information may lead to spurious correlation. New synthetic variables combine existing variables.

– Monthly mean temperature were converted into mean annual temperature and amplitude.

– Deep water swell energy flux was computed using swell period and significant height

$$E_f = \frac{1}{8}\rho g H_s^2 C_g \qquad with \qquad C_g = \frac{1}{2}g\frac{T}{2\pi} \tag{1}$$

Where $\rho$ is water density; Hs is significant wave height; Cg is wave group velocity; and T is wave period.

– Swell direction to the cliff.

### 2.4.4  Rock resistance inference

The database, filled with information from publications, results in more than 40 distinct lithological descriptions. We first grouped lithology into 9 groups with a similar classification to that of Woodroffe (2002) for historical comparison (Fig.9). But lithology alone does not governs rock mass mechanical properties. Tectonic inheritance, deformation, fracturing and weathering weaken the rock masses. Consequently, the rock constituting the cliffs are divided into rock mass strength criteria. Following
the practical examples from Hoek and Brown (1997), we propose to further aggregate Hoek and Brown's macroscopic rock mass strength categories into three categories.

Aggregation criteria are based on the fields lithology name, weathering, fracturing and comments, in which all published details on rock strength, structural geology, weathering were preserved. Rocks were classed into three resistance classes termed hard, medium and weak. One may note that a similar approach, but with two classes only, was adopted by the Eurosion project
consortium (Doody and Office for Official Publications of the European Communities, 2004). Hard rock cluster together granite, gneiss and limestones. Weak rocks are mainly poorly consolidated rocks (weakly cemented sandstones, glacial tills and glacial sands) or strongly weathered rocks. Weak rocks noticeably include well studied chalk cliffs. Medium resistant rocks correspond to claystone shales and siltstones.

## 3  Analysis / Results

### 3.1  Database content, completeness

The database is filled with 58 studies, out of which 47 are peer reviewed articles and 11 are public national databases, documenting 1530 cliff sites and 1680 erosion rate records. Indeed, some cliff sites where repeatedly measured over different periods. With more than 90% of complete fields, the database is rather well filled. However, the constitution of the database highlights





some generally lacking description. We mentioned previously the difficulty to find a description of cliff rock weathering and fracturing. Those fields are missing for 98.4% of records (corresponding to 53 publications).

## 3.2   Where was erosion measured?

Studies are mostly concentrated in Europe (42 studies, 1579 records), in Oceania, focused mainly on New Zealand (3 studies,
94 records) and Northern America (4 studies, 50 records). Asia (2 studies, 4 records) and South America (1 study, 1 record) are poorly represented. No literature was found for the entire African continent. Study locations are plotted in Fig.4.

## 3.3   How was erosion measured?

The number of studies is steadily growing since the middle 1990es (Fig.5), for every method types. Older studies exist and are present in Sunamura's database, however those papers were not available and/or cliff and measure description too poor to
be encoded in our database. The method most used method is the comparison of aerial photographs or historic maps, which correspond to a 2D method easy to apply and allowing erosion evaluation spanning several decades. 43 studies used this method representing 50% of published studies and 88% of the records. The second most used method is 3D type, which has become common from mid-2000. It represents 19 studies (22%) and 5% of records. Finally, some other methods are occasionally used. The 1D methods represent 8 studies (9%), 3.5% of the records.
Reported studies describe coastal processes along 20 m to 6,.4 km stretches of coastline. The median length is 600 m. Total survey duration vary from just 1 month to 7.1 ka, but half the data lie between 56 and 63 years given the bulk of aerial photograph comparison studies.

## 3.4   Examining relations between erosion rate and forcings

The purpose of the database is to examine the relationships between erosion rates, sites conditions and external forcing. Those
links were sought by means of exploration data analysis statistics.

### 3.4.1   Erosion vs rock mass properties

One of the first influent factor often pointed in literature is rock resistance (e.g., Benumof et al., 2000; Bezerra et al., 2011; Costa et al., 2004; May and Heeps, 1985). Figure 6 shows erosion rate distributions for the three rock resistance classes based on Hoek and Brown criterion. Three distinct behaviours can be seen. Hard rock erodes at a median rate of 2.9 $\mathrm{cm.yr}^{-1}$ with
a Median Absolute Deviation (MAD) of 3.4 $\mathrm{cm.yr}^{-1}$. Medium resistance rock coasts erode at around a median value of 10 $\mathrm{cm.yr}^{-1}$, with a MAD of 7.8 $\mathrm{cm.yr}^{-1}$. Due to the small number of observation of medium resistance rocks (63 observations), this resistance class should be considered carefully. Finally weak rocks erode at with a median value of 23 cm/yr and reach rates higher than 10 $\mathrm{m.yr}^{-1}$ with a MAD of 25 $\mathrm{cm.yr}^{-1}$.





Macroscopic rock mass strength classes, though possibly crude, exhibits the ordered behaviour expected by literature: weak rock erode faster than medium strength rock, and medium strength rocks erode faster than hard rocks. Central erosion rate values increase by a factor 2 to 3 from one class to the next.

These values are in agreement with Woodroffe's work (2002), but, even if those distributions are distinct, they are broadly
spread and multimodal.

## 3.5   Erosion vs marine forcings

In order to explore the influence of sea aggression, several variables were implemented in the database describing mean sea agitation and tidal range, and sea agitation during extreme events. All the variables concerning swell are strongly correlated. Hence, only three independent marine parameters are analysed in the following scatterplots (Fig.7): tidal range, wave energy
flux and number of storms.

All scatterplots appear to be widely spread and do not show simple linear relations. Indeed, Spearman's correlation coefficients, which evaluates monotonic relations between two variables, based on value ranks, are low. Furthermore, many tentative correlations cannot be trusted (p-value > 0.05). Those correlations and associated p-values are given in Table 1. Exploration of marine forcings indicate that none has an apparent effect on erosion rates, except a weak relation between tidal range and
erosion rates suggesting higher erosion for tidal ranges between 1 and 3 meters (yet not visible for medium resistant rocks).

## 3.6   Erosion vs climatic forcings

Concerning climatic forcings, recession rates are compared to temperature variation, frost frequency and amount of rainfalls. As for marine forcings, data is very scattered (Fig.8). Frost day frequency and rainfalls shows a positive trend with erosion rate for weak resistance rocks. Poorly consolidated rocks represents the large majority of type of rocks present in cold (>50
frost day per year) and rainy climates (> 1000 $\mathrm{mm.yr}^{-1}$) in the database. Only a few studies concern harder rocks under cold climate. However, even if a trend exists, data are really spread and Spearman's rank correlation coefficient is low (0.25 for frost, 0.07 for rainfalls) . Mean annual temperature do not show any clear correlation with erosion rate.

## 4   Discussion

### 4.1   Comparison to previous studies

The GlobR2C2 database provides a quantitative overview of current rocky coast erosion knowledge. This database is the first update since Sunamura (1992)'s seminal publication and adds 54 additional quantitative studies to the scientific debate. Its design allows to explore drivers of erosion. Historically, Woodroffe (2002) already tried to link erosion with lithology in a broadly reproduced graphic. This graph shows a clear pattern of increasing erosion rates with decreasing rock resistance. GloabR2C2 updates this classic graph using the same lithological classification (Fig.9). New knowledge does not change historical views,
but narrow down assumed erosion rate ranges both towards lower and higher rates. We also observe that supposed hard rocks





as granites or basalts can erode as quickly as 1 m.yr$^{-1}$. This is because resistance to erosion does not depend on lithological category alone, but also on the degree of weathering, jointing, folding etc. This graph, presented at a conference with sedimentologists triggered deep-hearted reactions for the lack of rock classification robustness in their community. This result confirms the choice for a less debatable rock resistance criterion instead of lithology. This geotechnical criterion is not perfect either.

It was inferred based upon authors' description of the cliff, thus it can include a part of interpretation and some degree of uncertainty.

## 4.2 What knowledge does GlobR2C2 compile?

The GlobR2C2 database is based on bibliographic references plus models and reanalysis used as proxies of forcings. Some bias are inherent to this kind of approach. The next paragraphs focus on different aspects of these limitations due to the use of

(i): erosion rate as a proxy of erosion, (ii) the use of model and reanalysis as proxy of forcing, (iii) the use of peer-reviewed journals.

### 4.2.1 Erosion rates, study duration and stochastic behaviour

Statistical exploratory data analysis (known as EDA) is a way to dissolve local particularity into a global analysis. Nonetheless, including every quantitative study implies mixing rates measured with different methods, accuracy, spatial and temporal

extents, which could be a source of bias. Erosion is a stochastic event: the fortuitous occurrence of a rare big event would influence the actual figure of the observed retreat rate. Rohmer and Dewez (2013) for instance, describe statistical indicators for testing the outlier nature of very large rock falls, with methods borrowed to hydrology, seismology and financial statistics. These indicators were applied to a chalk cliff site in Normandy (northern France) in Dewez et al. (2013). During the 2.5 years terrestrial lidar monitoring period, a massive 70'000 m$^3$ rock fall caused a local cliff top retreat of more than 19 m (Dewez et

al., 2013). That is more than one hundred years' worth of average retreat in one event. Estimated annual cliff recession rate rose from 0.13 m.yr$^{-1}$ to 0.94 m.yr$^{-1}$, a seven-fold increase, just by including this fortuitous, and definitely unrepresentative event (Dewez et al., 2013). Further demonstration is brought by other studies covering the same site. Costa et al. (2004) had estimated the recession rate to be ca. 0.15 m.yr$^{-1}$ in 29 years from aerial photos. And Regard et al. (2012), using millennial recession rates from 10Be accumulated in flint stones exposed in the chalk coastal platform, obtained 0.11 to 0.13 m.yr$^{-1}$ over

3'000 years.

  GlobR2C2 therefore addresses the concern of non-representative erosion values by compiling all studies available online, and retaining information from all sites and survey periods. In doing so, the actual dispersion of recession rate values is preserved and allows for recognizing outlying values (Fig.10).

### 4.2.2 Forcing proxies

While publication-derived cliff recession rates and cliff conditions could be forced into a coherent database framework, environmental forcings were so scarcely and heterogeneously documented that the same rationalization process was not possible on




the publication basis alone. Instead, publicly available global climatic and sea conditions database were used. These databases present the advantage of being spatially and temporally continuous thanks to reanalysed climate and sea state models. Their principal limitation is their coarse-grained definition compared to site specificities. Nevertheless, they document external forc-ings (i) in a uniform fashion (regular spatial and temporal sampling steps), (ii) for the entire globe, and (iii) reflect forcing

condition for durations spanning several decades. So, even if regional or continental data sets offer more resolved information in space or time, the global extent ensures that all cliff sites worldwide are documented uniformly.

### 4.2.3   Literature biases as future tracks to improve cliff evolution understanding

GlobR2C2's worldwide compilation shows that research in this domain is very active. A lot of quantitative data already exist. However, even if data coverage is somewhat global, publications turned out to focus mostly on a few western countries. This

finding is reflects the strategy of literature search adopted: only international and national literature published in English, French or Spanish were compiled. Due to the language barrier, we are aware that studies in Russian, German or Japanese languages, among others, were unwillingly obliterated.

Spatially, our search strategy did not flag scientific literature on the evolution of African and South American cliffs. Cliff recession studies are apparently focused on the richest areas where economically valuable coastal assets are exposed to losses.

This geographic distribution induces an over representation of temperate climates and a limited presence of some extreme climates or wave condition like equatorial or polar regions. Those extrema could nevertheless be a key for understanding effects of climate and wave conditions on cliff erosion.

Studies also focus on fast eroding coasts because they represents bigger risks and also because of methodological limitation. This fact biased the analysis by mostly documenting erosion distribution in higher values. The weight of this bias can be

approached thanks to the French CEREMA study. This study contains null erosion values for coastal sectors where the cliff was not seen to recess in a detectable manner on historical photographs. Yet this detection threshold is deemed to be of the order of $10\,\mathrm{cm.yr^{-1}}$ (Perherin et al., 2012), which is rather high, and null recession could reflect erosion situations anywhere in the spectrum between 0 and $10\,\mathrm{cm.yr^{-1}}$.

These null values represent 67% of the study rocky coasts, which means that low erosion rocky coasts are common and

ignoring this information can probably affect conclusions. In order to check the importance of the bias induced by those values, we explored two extreme cases. The erosion value was set to either a small value of $1\,\mathrm{mm.yr^{-1}}$ or to the detection threshold of $10\,\mathrm{cm.yr^{-1}}$ . Table 2 shows the influence of the null value in the distribution of erosion rate for the three Hoek and Brown rock strength classes. While the median and quantile absolute values are affected by the value attributed to null observations (TABLE), the expected order of rock sensitivity to erosion is maintained. Weak rocks erode at higher rates than

medium and hard rock. Therefore, we trust this result. Further, the dependency relationships flagged earlier remain. A weak positive correlation still exists between frost day frequency and a maximum tidal efficiency for tidal range between 1 and 3 m still is observed.



## 5 Conclusions

Compared to inshore cliffs, coastal cliffs obviously erode quicker because of the sea presence. The GlobR2C2 v1.0 database compiles ca. 2000 rocky coast cliff retreat data from an online global literature search published before 2016. It is the first attempt of this kind since Sunamura's seminal publication in 1992. The investigated period adds information arising from

the quantitative revolution of lidar technology, structure-from-motion technique, accessible to scientists with little background in photogrammetry and massive release of aerial photographic archives of mapping agencies from western countries. The data compiled in GlobR2C2 is heterogeneously distributed in terms of retreat rates, geographical location, cliff nature and climate settings. Even if further research should aim at completing little studied geomorphic contexts of the globe, existing information clearly shows that cliff retreat is most clearly governed by the lithological nature of the cliffs. The dependence of

cliff recession rates on rock types is best expressed using a geotechnical parameter, the Hoek and Brown (1997) macroscopic rock mass strength parameter. Rocks classed as weak (recession rate median: 23 $\mathrm{cm.yr^{-1}}$) erodes 2-3 times faster than medium strength rocks (median rate: 10$\mathrm{cm.yr^{-1}}$), themselves erode 2-3 times faster than hard rocks (median rate: 2.9 $\mathrm{cm.yr^{-1}}$). Using solely a lithology denomination in the way of Woodroffe (2002) historical graph (Fig.9), lithologic types exhibit a similarly ordered behaviour (Fig.6), even if geologists contest the robustness of these denominations as proxies for rock strength.

Together with cliff settings compiled from publications, GlobR2C2 also records continental climate and marine conditions at study sites from reanalysed models for their global, spatial and temporal sampling regularity. Both forcings exhibit weak relations with cliff recession rates. In relative terms, however, climate (i.e. frost days frequency) exhibits a stronger influence than marine forcing. Influence of the sea is only slightly visible in this dataset through a maximum efficiency of erosion for tidal ranges between 1 and 3 meters.

Our data divides into three classes of resistance, following the Hoek and Brown parameter. The most resisting (respectively least resisting) rocks are found to lead to retreat rates less than 0.1 $\mathrm{m.yr^{-1}}$ (Q83) (respectively up to 85 cm). Medium-resistance rocks are not studied enough to give a precise range of retreat rates. Climate seems to be more efficient and frost seems to have the strongest influence.

We conclude at this stage that rocky coast erosion is primarily driven by cliff settings with second-order but non-negligible

modulations by marine and continental forcings (Fig.2). These findings are of primary interest for coastal erosion models which, up to now, focus mostly on marine forcing (e.g., Anderson et al., 1999; Trenhaile, 2000).

*Code availability.* TEXT

*Data availability.* TEXT

*Code and data availability.* TEXT

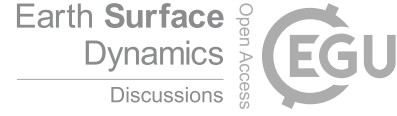

# Appendix A

## A1

*Author contributions.* TEXT

*Competing interests.* TEXT

5 *Disclaimer.* TEXT

*Acknowledgements.* MP's PhD fellowship was funded in equal part between BRGM, the French geological survey, and French Région Midi-Pyrénées under grant number. L. Roblou is warmly acknowledged for advices about tide calculation. We thank S. Carretier, C. Garnier E. Nardin, D. Rouby for their support and wise advices during PhD committees.





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





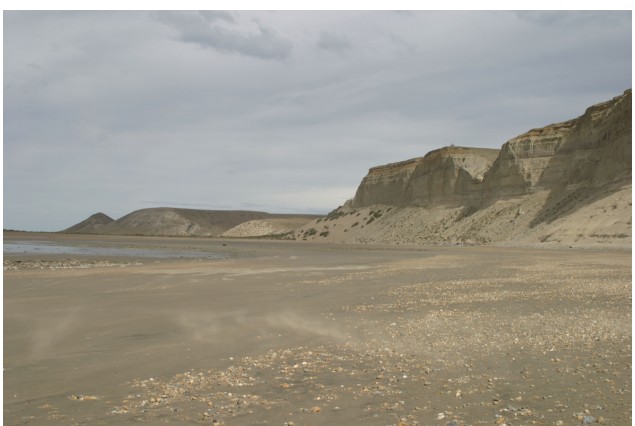

**Figure 1.** Evidence of sea driving coastal cliff erosion. The cliff in the foreground is similar than that in the background except that the one in the background has been protected from the sea by a sand spit. Obviously, the cliff with sea at its base retreats faster (the cliff face is more vertical). Photo from Punta Quilla, Patagonia, Argentina.




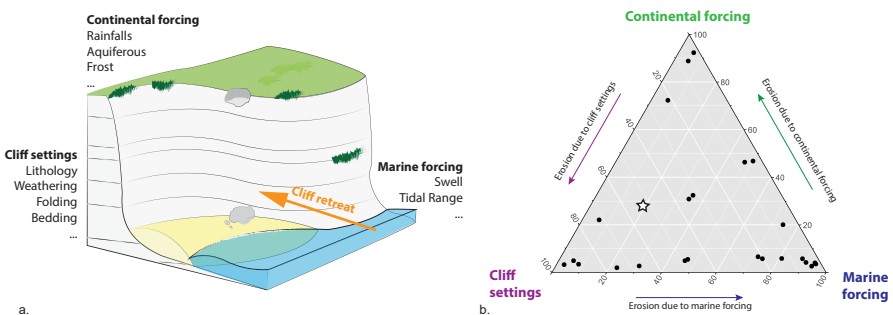

**Figure 2.** a. Scheme of rocky cliff erosion drivers. b. Relative cliff retreat drivers reported from published litterature. Eroding factors are grouped within three main classes: (i) "marine forcing", (ii) "continental forcing" encompassing weather condition and continental groundwater, and (iii) "cliff settings". Authors point of view is summarised as a percentage of those three forcing based on abstract content. The star positions the result of the present study.





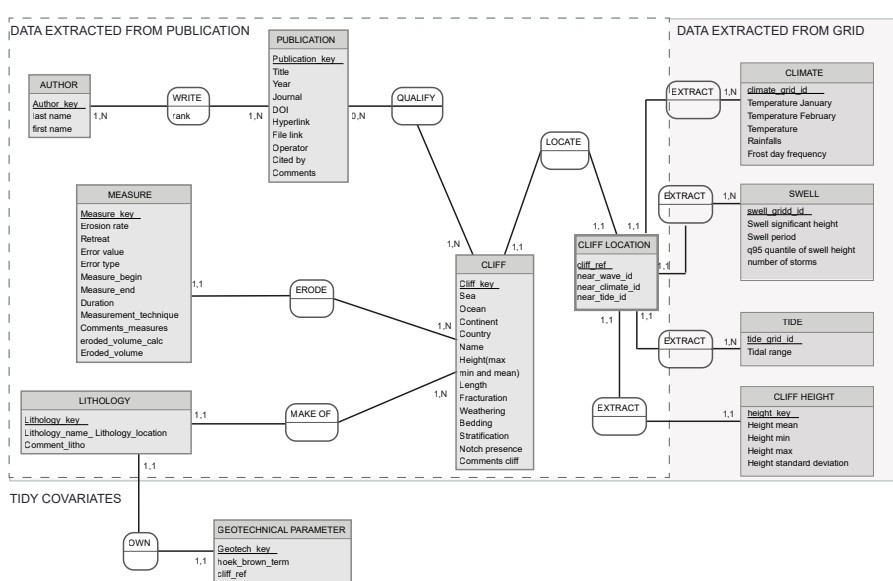

**Figure 3.** Conceptual data model of cliff erosion database globr2c2. Primary key are underlined and numbers are cardinalities



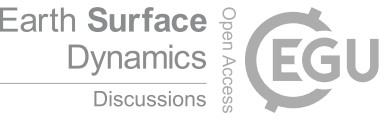

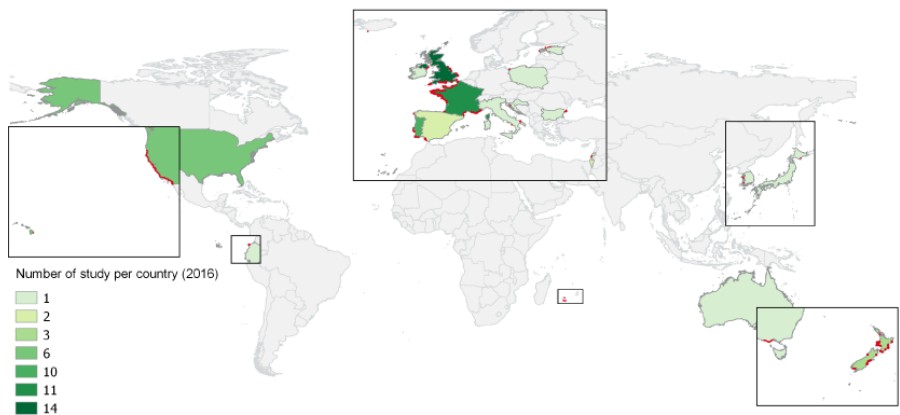

**Figure 4.** Cliff site locations (red dots) and number of studies by country contained in the database GlobR2C2 (publication before 2016)



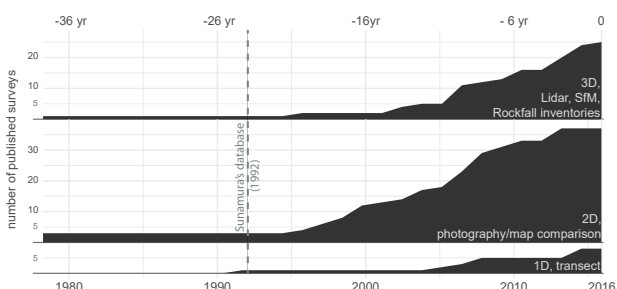

**Figure 5.** Number of rocky cost erosion studies per different method trough time





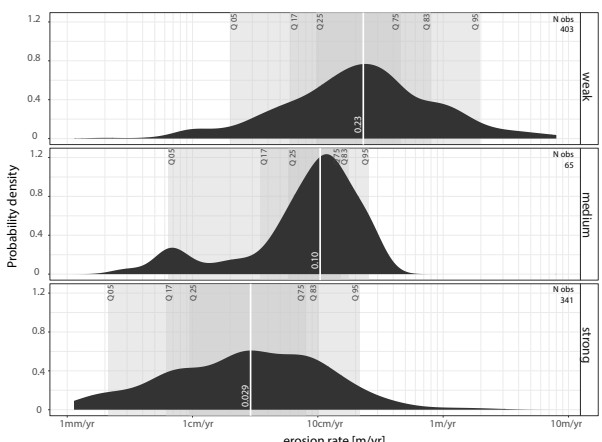

**Figure 6.** Erosion rate distribution for each one of the rock resistance class. Those resistance classes were attributed according to a simplified hoek and brown rock mass strength criterion merging lithological description and fracturing/weathering state of the rock.



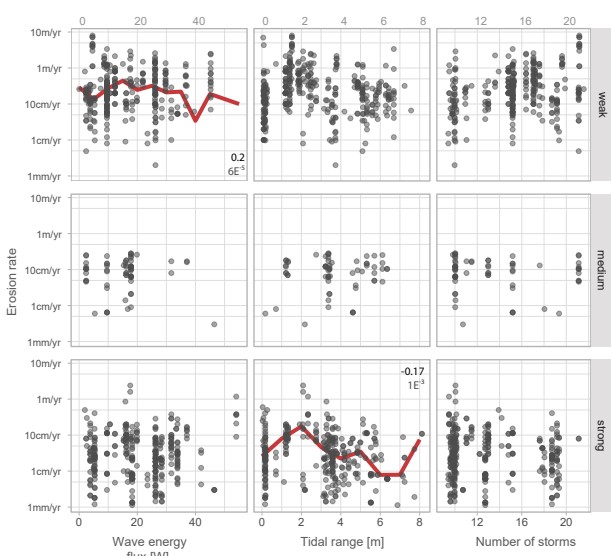

**Figure 7.** Erosion rate vs marine forcings (wave energy flux [w], tidal range [m] and number of storms) for each one of the hoek-brown rock resistance class. Lines beneath scatterplots represents moving median per bin and numbers are Spearman's correlation coefficient. They were only repesented when p-value was significant





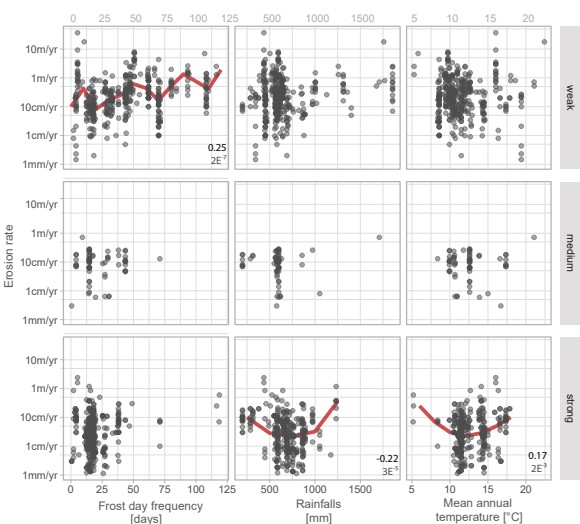

**Figure 8.** Erosion rate vs climate forcings (frost day frequency [days], annual cumulated rainfalls [mm] mean annual temperature []C) for each one of the hoek-brown rock resistance class. Lines beneath scatterplots represents moving median per bin and numbers are Spearman's correlation coefficient. They were only repesented when p-value was significant





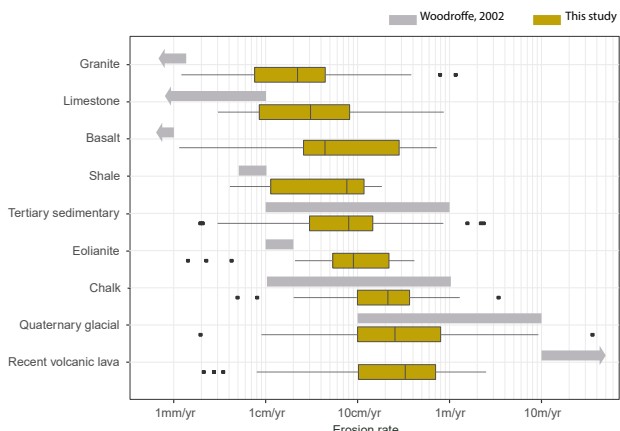

**Figure 9.** Range of erosion rate within different lithology. Comparison between woodroffe's 2002 study and this one.



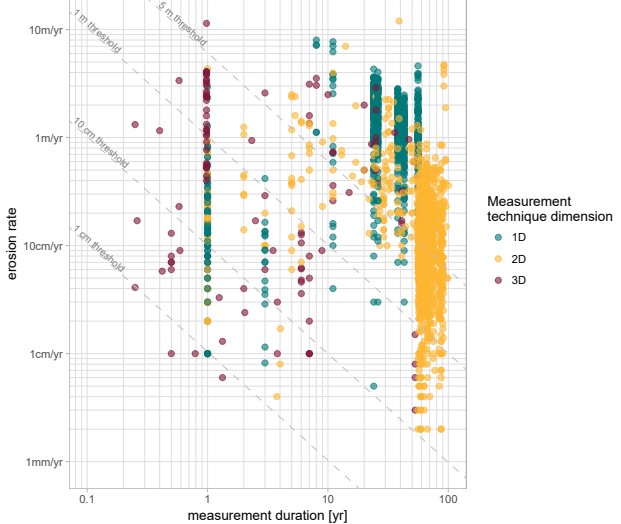

**Figure 10.** Survey time vs erosion rate by groups of measuremant techniques.

**Table 1.** Influence of value assignation for CEREMA estimates under the threshold $(0.1\ \mathrm{m.yr^{-1}})$ on erosion distribution for different strength classes

| | weak | | medium | | hard | |
|---|---|---|---|---|---|---|
| | corr | pval | corr | pval | corr | pval |
| tidal range | -0.23 | 2.74E-08 | -0.02 | 8.53E-01 | -0.24 | 1.76E-15 |
| mean swell height | -0.16 | 1.11E-04 | 0.00 | 9.81E-01 | -0.09 | 1.92E-03 |
| number of storms | 0.33 | 9.42E-16 | -0.20 | 1.04E-01 | 0.11 | 3.85E-04 |
| rainfalls | -0.27 | 4.41E-11 | -0.09 | 4.81E-01 | -0.19 | 1.40E-10 |
| frost day frequency | 0.43 | 4.17E-26 | 0.04 | 7.61E-01 | -0.06 | 4.75E-02 |
| mean temperature | -0.18 | 1.53E-05 | -0.02 | 8.79E-01 | 0.18 | 1.14E-09 |
| wave energy flux | -0.16 | 2.25E-04 | 0.04 | 7.65E-01 | -0.07 | 1.47E-02 |
| mean swell period | 0.03 | 4.74E-01 | 0.08 | 5.13E-01 | 0.08 | 1.14E-02 |
| q95 swell height | -0.34 | 2.31E-16 | -0.02 | 8.70E-01 | -0.15 | 7.02E-07 |
| q95 swell period | -0.10 | 2.39E-02 | 0.11 | 3.86E-01 | 0.05 | 1.12E-01 |

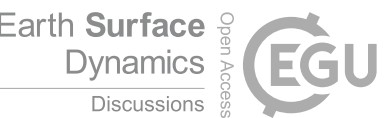

**Table 2.** Influence of value assignation for CEREMA estimates under the threshold ($0.1$ m.yr$^{-1}$) on erosion distribution for different strength classes

| | weak | | | | | medium | | | | | hard | | | | |
|---|---|---|---|---|---|---|---|---|---|---|---|---|---|---|---|
| | med | q5 | q27 | q83 | q95 | med | q5 | q27 | q83 | q95 | med | q5 | q27 | q83 | q95 |
| without null | 0.23 | 0.018 | 0.1 | 0.85 | 2.499 | 0.104 | 0.006 | 0.063 | 0.18 | 0.269 | 0.029 | 0.002 | 0.01 | 0.106 | 0.286 |
| null = 0.01 | 0.129 | 0.001 | 0.006 | 0.683 | 1.806 | 0.102 | 0.002 | 0.049 | 0.18 | 0.269 | 0.001 | 0.001 | 0.001 | 0.024 | 0.112 |
| null = 0.1 | 0.129 | 0.01 | 0.1 | 0.683 | 1.806 | 0.102 | 0.006 | 0.063 | 0.18 | 0.269 | 0.1 | 0.006 | 0.1 | 0.1 | 0.112 |