# Peer review of "GlobR2C2 (Global Recession Rates of Coastal Cliffs): a global relational database to investigate coastal rocky cliff erosion rates variations."

_Earth Surface Dynamics, 2018_

## Referee Comment (RC1) · L. Naylor (Referee) · 29 Mar 2018

General Comments

This paper presents a much improved, extended and comprehensive database of global coastal cliff erosion which brings together the rapidly expanding number of papers in this field, in a rigorous and comprehensive analysis. Of particular import is that their analysis allows for improved understanding of the importance of rock resistance over lithology or climatic parameters as the key factor controlling erosion rates. They found that rock mass properties like joints and fractures are a fundamental control of coastal rock cliff erosion rates. They have come to these conclusions through creating a thorough, rigorous and repeatable database that can be extended through time as

more papers are published. It is thus of great scientific importance and serves as a valuable tool which can be built upon. The authors are to be highly commended for their efforts.

I really enjoyed reading this manuscript and once a moderate level of corrections are made, this will make a superb and much needed contribution to the literature. The key areas for improving this manuscript lie in six areas:

a) Improving the explanation of your methods and statistical analyses. Why did you use the tests you did? Which aspects did you test statistically?

b) Considering 'dating techniques' as an additional category of how erosion is measured and adding this category into your analysis (eg Figure 5) or explaining where this content best fits in your classification from 1D to 3D studies.

c) Threshold, non-linear behaviour of coastal rock cliff erosion. Many of the types of cliffs included often display threshold-driven, non-linear behaviour. Whilst I appreciate you needed to standardise your reporting of erosion to mm/yr-1, I also wonder if it is possible to evaluate the degree of stochasticity /non-linearity in the database. For example, it may be that certain rock resistance types are more prone to non-linear, stochastic erosion events or that the temporal frequency between erosion events varies by rock resistance category or another parameter. Finding a clever way including this alongside your mm/yr-1 would improve awareness of the behaviour of these systems for risk managers, hazard scientists and geomorphologists alike.

d) Wider context. In places, the analysis and discussion of this paper is too narrowly focussed on coastal rock cliff erosion, rather than drawing on evidence from recent shore platform research which displays similar trends around the importance of geological contingency, the importance of rock mass properties and weathering/rock breakdown (bio/chem/phys) processes helping prepare rock coast landforms for erosion. This includes the early conceptual models of cliff erosion by Sunamura as well as recent papers on rocky shore platforms.

e) Figure 1 and your discussion of it shows the importance of the wider geomorphic context in influencing erosion rates. This does not appear to be taken into consideration in the current version of your model. It would be useful for the authors to explore how this may be possible, so that a global analysis of how submarine to cliff-top coastal landforms vary around the globe and how this affects erosion rates. For example, what proportion of cliffs globally are currently shielded by offshore features such as those in part of Figure 1? Does this vary by rock resistance of the cliffs or are other factors influencing this? I realise that much of this may be beyond the scope of your current paper, but it may be useful to signpost this in your current paper, perhaps using data from both parts of Figure 1 as an example to illustrate how cliff erosion rates are modified by their wider geomorphic context, and thus are partly geomorphologically controlled.

f) Lastly, it would be useful to signpost the wider significance of your work for coastal hazards scientists, geologists and in the context of changing storminess and sea level rise. It also would be helpful to highlight the potential to extend the database to include shore platform erosion rates. This would help show the wider relevance and import of your work.

Specific Comments (SC), Technical Comments (TC)

Title

SC - You may wish to change the title to better capture the global database /analysis that is, to me, a significant strength of your paper and a very strong addition to the literature.

Abstract

SC - Show the wider relevance of your important work here

Introduction:

TC - First sentence needs reference and second sentence needs a direct quotation.

SP – wave-cut vs shore platform needs a little more discussion

TC – para 25 Fig 3 or Fig 5?

TC – para 30 cite Viles 2017 Geomorphology

Method:

SC – define your boundary conditions and cite Kennedy who first used this term explicitly in rock coast geomorphology

SC – systematic search method needs improving, this can either be quite simple as per Figure 1 in Naylor et al. 2010 or following the more detailed PRISMA method (Moher et al. 2015) stemming from medical science.

TC – Merise needs a year , pg. 3 para 25

SP – pg 4, Para 10 sentence 1 examples adding would be helpful to aid understanding of your database.

TC – pg, 4 para 5, first sentence could be reworked

SP – section 2.3.1. a) Only English is mentioned here but Spanish and French is mentioned earlier. B) define your search method and strings (perhaps as supplementary material), this will make this part of your work reproducible and improve rigour.

SP – 2.3.4 add Hurst et al. 2017 as reference for 1000s of years scale

SP - 2.3.5 last sentence is unfinished

SP – 2.3.9 can you validate your assertion in the last sentence?

SP – 2.4.4 Not all of your core readers will be familiar with the Hoek Brown criterion as it is a geotechnical/engineering criteria. I recommend you add some background information and some rationale for why this was the best metric to use. Here it would be good to explain why Selby 1980 is less suitable than Hoek Brown.

Section 3:

SP – 3.3 See comment above about dating methods.

SP – 3.4.1 fewer medium resistance rock studies, perhaps make this as a suggestion for future research in your conclusions, along with the present geographic limitations?

Section 4: SC- Para 5, page 11 - more detail on this conference, a specific pers comm would help here too.

SC- Weathering, jointing, discontinuities – Sunamura included these parameters in his early conceptual models of rock cliff, rock coast and shore platform erosion, showing how they contributed to the reducing the resisting force of rocks. The influence of these on erosion processes and rates has been more recently discussed for rocky shore platforms (See Cruslock et al. 2010, Naylor and Stephenson, 2011, Stephenson and Naylor 2012) and biology (Naylor et al. 2012).

SC - 4.2.1 para 20, this is where the threshold, non-linearity comment above relates.

SC - 4.2.3 pg 12, para 10, I recommend you refer to Kennedy et al. 2014 here as this volume has no chapter on Africa, which accords with your analysis of rocky cliffs. Doing so would strengthen this point.

SP – pg 12, para 20, does this mean it relates only to softer rocks? Please clarify. TC - Pg 12, para 25, I think this is table 2?

SC - Page 13, para 25 there are many newer rock coast evolution models including consideration of the impacts of climate change (e.g. Limber, Ashton, Trenhaile) that are worth looking at to improve your link to modelling.

Technical comments for the whole the Manuscript:

Minor improvements to your English is needed occasionally throughout the manuscript Measure often needs to be measurement

Page 13 – inshore could be confused with 'inshore waves'; I recommend using terrestrial instead.

Page 20, what does Q83 refer to? Also add a final sentence, or extension to it that shows which rock categories this relates to.

In a few places you talk about rocky coast erosion, your topic is coastal rocky cliff erosion. For clarity about your scope and the contents of your paper, the latter term should be used throughout.

---

## Referee Comment (RC2) · C. Moses (Referee) · 29 Mar 2018

Overview comments: The paper seeks to explain variations in sea cliff erosion rates, using a global database populated by cliff erosion rate data derived from scientific literature and national databases up to 2016. Marine and climate forcing factors are derived from models and data reanalysis in order to provide a uniformity of approach. Sea cliff lithological factors are characterised using the Hoek and Brown (1997) classification system, again in order to provide a uniform approach, and cliff height is been extracted from the 8" global DEM. The paper represents the most comprehensive collation and analyses of rock coast erosion data to date and is scientifically important in two key respects. First, it provides analyses and insights into key factors controlling rock coast erosion rates on a global scale. Second, it illustrates limitations of existing

studies/current gaps in knowledge in assessing the relative importance of lithological, subaerial and marine forcing factors. In so doing, it helps to set a new research agenda for the study of rock coast erosion dynamics and this could usefully be made clearer in the paper.

The conclusion is that rock resistance, rather than rock type per se, is a key influencing factor and that the number of frost days influence the erosion rates of only weak rock sea cliffs. Rainfall amount and marine forcing factors show no significant relationships with cliff erosion rates. This is interesting in that there is a keen debate on the importance of subaerial (weathering) versus marine forcing factors in the development of rock shore platforms, which are an integral component of the rock coast system. This debate extends also to cliffs. For example, it is known on the Chalk of SE England that most rockfalls occur during the winter (May, 1971; Hutchinson, 1972) associated with increased rainfall and lower temperatures. Lawrence et al. (2013) assess the contribution of sea water weakening to chalk cliff instability and Lageat et al. (2006) and Henaff et al. (2002) assess the influence of elevated groundwater and rock saturation associated with long periods of antecedent rainfall. Although this study assesses cliff erosion rates in relation to temperature variation, frost frequency and amount of rainfall, it would be interesting to give some consideration to duration of rainfall (as a proxy for degree of rock saturation) to see if this is important.

More specific comments: Page 1 Line 3: 'It turns into variable erosion rates' suggest amending this to 'Cliff erosion rates are highly variable over 4 orders . . . . . .' in order to improve clarity. Are these figures from the database? If so, it may be better to give the variation in rates after describing the database. Line 6: – it would be helpful to be clear about what is meant by erosion rate – rate of cliff-top retreat, volume of material removed? Is GlobR2C2 populated entirely with erosion rate data from publications? How is the Cerema national database incorporated? There is mention in the paper of the Eurosion database – is this also incorporated into GlobR2C2? The Eurosion database is being updated and extended by the Emodnet Geology project

and so there are new data, that the authors may wish to investigate, available at http://www.emodnet-geology.eu/data-products/ (coastal behaviour). I am wondering if the title is an accurate reflection of the database if it incorporates more than the scientific published literature. Line 12: space between numerical value and SI symbol (throughout). Line 13: Sentence beginning 'every other relations. . ..' Could be recast to improve clarity.

Line 18:. . . fundamental driver – suggest adding 'of cliff retreat'. Line 19: Remove " after limited. It would be helpful, in the introduction, to provide more context on the role of rock shore platforms in the dynamics of coastal rock cliff erosion dynamics. Although shore platforms are mentioned it would be helpful, for readers not familiar with the rock coast system, to set the context by outlining all of the key components. For example, Fig. 2 could usefully show the shore platform. Page 2 Line 12: Sentence beginning 'Climate through. . .' remove the s from precipitations; prepare for it? Fig. 2 is referred to on line 16 and Fig. 5 on line 29 – Figs. 3 and 4 are not mentioned – refer to Figs in order throughout. Par beginning line 19: 'they are inconclusive because. . .' it would be helpful to have more context on the focus of these papers as they did not necessarily set out to analyse the contribution of each factor etc., perhaps due to data limitations? Par beginning line 29: it would be helpful to have some more detail on the type of study – what they measure, degree of accuracy, limitations etc. (historical maps, air photos, TLS, Lidar, photogrammetry, use of drones). Page 3 Line 1: 'high time resolution of up to 20 minutes' – it would be helpful to say what this high temporal resolution data records – removal of individual small rock fragments from the cliff face? Line 5: 'study their relative efficiency'- not clear how this relates to linking erosion rates and external forcings – perhaps amend sentence to improve clarity. Line 8: 'reduces information to the largest common denominator' – yes, this may be a limitation but it is also an opportunity! It would be helpful if the paper can set out, on the basis of this study, a clear statement of the scale/resolution of study and also the important factors to record for future studies of rock coast erosion – in order to improve the resolution of the GlobR2C2 in the future.

Line 18-19: it woud be helpful to say here what databases are used. Line 27-30: sentence beginning 'It helps. . .' and the next sentence could be made clearer. For example, I am wondering if the conceptual exercise really minimises data capture? Should it be 'maximise data capture and minimise data redundancy'? Page 4 Line 8: three types of sources?; are the data from scientific papers really raw data? Not clear what is meant by gridded data and tidy covariates. Par beginning line 9: this could usefully be expanded to aid explanation. For example, is the method of measuring cliff erosion recorded and the time period over which it is measured? Figures will need to be re-numbered in order to ensure that they are referred to in the correct order. Section 2.3.2 Cliff lithology and description: it would be interesting to know how you have dealt with composite cliffs in the database – for example, a composite cliff may contain materials of different hardness/resistance at the toe and so marine forcing may be of reduced importance in such cases. Page 5 Line 2 – 4: meaning unclear and it would be helpful to recast these two sentences to improve clarity. Line 7: not clear what is meant by 'a primary key'. Line 10: etc. – please specify what is included in the etc.! Line 14: suggest amending to 'estimates. . . of volume loss to precise measurements using, for example, lidar. . ..' Line 15: suggest amending to '(iii) spatial extent along the coast. . ..' Line 23: not clear what is meant by 'the oldest method is rockfall inventory' Line 29: suggest amending to 'but with two caveats' Line 31: it would be helpful to say how data were 'specifically treated' beforehand in order to prevent bias. Page 6 Line 3: is it the case that faster eroding cliffs are more often sampled – are more densely populated cliffs not also more often sampled by regional/national authorities? Line 7: suggest amending to '. . .that quality of photographs limits. . ..' Line 11: not clear what is meant by 'and produce wetting drying cycles' – does this mean, influences the vertical extent of wetting drying cycles on the cliff face? How about any potential influence on groundwater levels in more porous rocks? Line 13-14: it would be helpful to add some explanation to the harmonics. Line 29: time steps Line 30: spelling – below. Page 7 Line 17: 'thus, 3D measures. . .' (rather than this?) Page 8 Line 12: Fig. 9 is referred to but the last Fig referred to was Fig 5 – Figs 6, 7 and 8?
Section 2.4.4: it would be helpful to have some more contextual detail on the Hoek and Brown rock resistance classification that is used in the study. Page 9. Line 6: Fig. 4 is out of synch. Line 8: suggest amend to '. . .1990s. . .for every type of method' Line 15: 6.4 km Line 26: provide the number of observations for each class rather than just one. Page 10 Line 17: '. . .amount of rainfall.' Line 27: '. . .design allows an assessment of the drivers of erosion'? Page 11 Section 4.2.1 See also Michoud et al. (2012) who estimated cliff retreat of the "Dieppe landslide": 'activated on 17–18 December 2012. . .. . . we measure a cliff retreat up to 40 m along two active scarps over 70 m wide' (p. 415). Page 12 Line 10: 'this finding reflects' (remove is) Line 29: amend TABLE Page 13 It would be helpful to have some discussion of the importance of weathering that can be drawn on for the conclusion. It would also be helpful to make some recommendations for future studies of rock coast erosion that would help to address the data gaps identified in the compilation of GlobR2C2. Figures Figure 1: suggest amend to: '. . .is similar to that. . .' Figure 2: diagram a could usefully show the shore platform; there is no mention of faulting in the cliff settings – if it is included then it would be helpful to mention it; not clear what is meant by 'aquiferous' in the continental forcing. Diagram b seems to use only half of the 58 studies that are used in the database (there are $\sim$ 23 dots on the graph). Also, it is not clear what is meant by the 'authors point of view'. It would be helpful to have some more explanation either in the caption or in the text. Figure 6: Hoek and Brown Figure 8: typo after temperature Figure 9: Woodroffe

Henaff, A., Lageat, Y., Costa, S., Plessis, E., 2002b. Modalités du recul des falaises du Pays de Caux. In: Delahaye, D., Levoy, F., Maqaire, O. (Eds.), Geomorphology: From Expert Opinion to Modelling. A Tribute to Professor Jea-Claude Flageollet, Strasbourg, pp. 225–233.Hutchinson, 1972 Lageat, Y., Hénaff, A., Costa, S., 2006. The retreat of the chalk cliffs of the Pays-de-Caux (France): erosion processes and patterns. Zeitschrift für Geomorphologie 144, 183–197. Lawrence, J.A., Mortimore, R.N., Stone, K.J. and Busby, J.P., 2013. Sea saltwater weakening of chalk and the impact on cliff instability. Geomorphology, 191, pp.14-22. May, V., 1971. The retreat of chalk

cliffs. The Geographical Journal 137, 203–206. Michoud, C., Carrea, D., Costa, S., Derron, M.H., Jaboyedoff, M., Delacourt, C., Maquaire, O., Letortu, P. and Davidson, R., 2015. Landslide detection and monitoring capability of boat-based mobile laser scanning along Dieppe coastal cliffs, Normandy. Landslides, 12(2), pp.403-418.

---

## Author Comment (AC1) · 11 May 2018

We wish to thank Cherith Moses (reviewer 2) for reviewing this manuscript and help us improve it.

**Overview comments: The paper seeks to explain variations in sea cliff erosion rates, using a global database populated by cliff erosion rate data derived from scientific literature and national databases up to 2016. Marine and climate forcing factors are derived from models and data reanalysis in order to provide a uniformity of approach. Sea cliff lithological factors are characterised using the Hoek and Brown (1997) classification system, again in order to provide a uniform approach, and cliff height is been extracted from the 8" global DEM. The paper**

**represents the most comprehensive collation and analyses of rock coast erosion data to date and is scientifically important in two key respects. First, it provides analyses and insights into key factors controlling rock coast erosion rates on a global scale. Second, it illustrates limitations of existing studies/current gaps in knowledge in assessing the relative importance of lithological, subaerial and marine forcing factors. In so doing, it helps to set a new research agenda for the study of rock coast erosion dynamics and this could usefully be made clearer in the paper.**

To best illustrate limitations of existing studies we added a last paragraph in the discussion: 4.2.5. Toward a new rocky coast cliff research agenda This bibliographic synthesis has highlighted the strengths and weaknesses of the current rocky coast research efforts. The last three decades's trend has gone towards increasing the quality and the resolution of cliff recession data and on documenting growing number of sites; which is good. What this study highlights however is a lack of description of critically useful parameters to understand cliff evolution dynamics: (i) cliff height; (ii) finer rock mass characteristics description, in particular weakening phenomena such as weathering and fracturing; and (iii) foreshore description, in particular its type (sand beach/pebble beach/rock platform) and geometry (elevation, slope, width). Moreover, the geographical distribution of studied sites highlight a major gap of knowledge under extreme climates (tropical, equatorial and glacial) or for slowly retreating cliffs. We also found that literature concerned with cliff retreat was not simultaneously trying to link shore platform processes to cliff retreat or how local variations affected cliff retreat specifically.

**The conclusion is that rock resistance, rather than rock type per see, is a key influencing factor and that the number of frost days influence the erosion rates of only weak rock sea cliffs. Rainfall amount and marine forcing factors show no significant relationships with cliff erosion rates. This is interesting in that there is a keen debate on the importance of subaerial (weathering) versus marine forc-**

**ing factors in the development of rock shore platforms, which are an integral component of the rock coast system. This debate extends also to cliffs. For example, it is known on the Chalk of SE England that most rockfalls occur during the winter (May, 1971; Hutchinson, 1972) associated with increased rainfall and lower temperatures. Lawrence et al. (2013) assess the contribution of sea water weakening to chalk cliff instability and Lageat et al. (2006) and Henaff et al. (2002) assess the influence of elevated groundwater and rock saturation associated with long periods of antecedent rainfall. Although this study assesses cliff erosion rates in relation to temperature variation, frost frequency and amount of rainfall, it would be interesting to give some consideration to duration of rainfall (as a proxy for degree of rock saturation) to see if this is important.**

We agree that rock saturation may be an important phenomenon. Following your remark, we investigated how it could be characterized effectively. Rock saturation is a combination of two parameters: rock reservoir capacity and efficiency of rainfalls to load the aquifer. We explored several ways to characterise those two parameters but could not work them out in publishable state within the imparted correction deadline. An element worth noting is that our approach aims at grasping the general trends rather than explaining specific collapse events. Nevertheless, your remark is very valuable and is one of the strands towards which GlobR2C2 will be extended in the future. You will find in the following lines the state we reached in this endeavour.

- For rainfall: the main issue was to find a rainfall data set suitable for such an analysis. The putative dataset needs to propose a chronicle of rainfall with a fine enough temporal (e.g. daily or hourly) and spatial (tens of kilometres at the worst) sampling step at global scale starting from the mid-20th century. The closest we found, and well short of these requirements, was NASA's TRMM (Tropical Rainfall Measuring Mission) rainfalls data set, which spans 17 years of record (1997-2015) for grid cells of $0.25° \times 0.25°$ at 7 hours time step between $50°N$ and $50°S$. We would be delighted to hear about other data sources. And once the data set

is flagged, there'd still be a need to flag the appropriate information parameter to describe the rainfall regimes in a discriminant way to characterise aquifer-loading efficiency from rainfall. This is currently beyond our field of expertise. Exploratory tests with the global Koeppen-Geiger climate zone classification was not fine enough to discriminate cliff retreat sites.

- For rock capacity characterisation. We identified the GLHYMPS global database of Gleeson et al., 2014. This database maps at global scale values of porosity and permeability. We need further investigation and bibliographic work to find a way to integrate those values in the database. We seek to identify a well-established macroscopic parameter similar to the Hoek and Brown rock mass rating index with classes that would characterise if the rock would behave as a reservoir or not. Again, this goes beyond the initial scope of our paper.

**More specific comments: Page 1 Line 3: 'It turns into variable erosion rates' suggest amending this to 'Cliff erosion rates are highly variable over 4 orders' in order to improve clarity.**

The amendment was done.

**Are these figures from the database? If so, it may be better to give the variation in rates after describing the database.**

These figures were already known from Sunamura's work in 1992. We appended this information in the abstract.

**Line 6: it would be helpful to be clear about what is meant by erosion rate, rate of cliff-top retreat, volume of material removed? Is GlobR2C2 populated entirely with erosion rate data from publications? How is the Cerema national database incorporated? There is mention in the paper of the Eurosion database – is this also incorporated into GlobR2C2? The Eurosion database is being updated and extended by the Emodnet Geology project and so there are new data, that the au-**

**thors may wish to investigate, available at http://www.emodnet-geology.eu/data-products/ (coastal behaviour). I am wondering if the title is an accurate reflection of the database if it incorporates more than the scientific published literature.**

Several items are contained in this remark. Let us review them on by one:

**Erosion rate.** In the abstract, the mention of this term was completed like this "...erosion rates from publications, ..." to clarify as required. In the main text, we use cliff retreat rate when appropriate. The database aggregates cliff evolution quantities from a variety of methods and practices. The database records both, the original quantitative information, with a name labelling the method used. We then grouped the various methods in 1D, 2D, 3D higher-order measurement categories. Section 2.3.4 Measure description is dedicated to document this process. We think that the generality of "cliff retreat rate" is sufficiently encompassing to be used in the abstract without entering in any technicality.

**CEREMA database.** The Cerema national database incorporation is described into a dedicated section (2.3.5). For more clarity we added its specific contribution to the data set. "The French CEREMA institute published a systematic national coastal cliff recession inventory Perherin et al., 2012) based on aerial photograph comparison every 200 meters stretch of cliff along the entire French metropolitan coastline (1800 km of coastal rocky cliffs, it correspond to 465 (53%) values in the database).)"

**Emodnet and Eurosion.** We did not use the "data products" because the original data is hidden behind a complex processing. On the contrary, we use available local case studies from Eurosion when the completeness of the data is enough for our purpose .

**Line 12: space between numerical value and SI symbol (throughout).**

Spaces between numerical value and SI symbols have been checked all along the document

**Line 13: Sentence beginning 'every other relations.' Could be recast to improve**
**clarity.**

It has been rephrased

**Line 18: fundamental driver suggest adding 'of cliff retreat'.**

The amendment was done.

**Line 19: Remove " after limited.**

It is a mistake, we forgot the opening ", we put it before "our understanding

**It would be helpful, in the introduction, to provide more context on the role of rock shore platforms in the dynamics of coastal rock cliff erosion dynamics. Although shore platforms are mentioned it would be helpful, for readers not familiar with the rock coast system, to set the context by outlining all of the key components. For example, Fig. 2 could usefully show the shore platform.**

We think the shore platform is already present in our description of the processes leading to cliff erosion. To make it clearer to both reviewer we improved the figure 2 and added some sentences within the introduction to better articulate between the shore platform and rock coast erosion.

**Page 2**

**Line 12: Sentence beginning 'Climate through remove the s from precipitations; prepare for it?**

The amendment was done.

**Fig. 2 is referred to on line 16 and Fig. 5 on line 29 – Figs. 3 and 4 are not mentioned refer to Figs in order throughout.**

We modified figures numbering to follow the sequential order.

**Par beginning line 19: 'they are inconclusive because it would be helpful to have more context on the focus of these papers as they did not necessarily set out to**

**analyse the contribution of each factor etc., perhaps due to data limitations?**

We add more context and consequently this part changes to: "Those studies are often risk management (Gibb, 1978; Hapke et al., 2009). Or they can be focused on a certain type of rock to understand cliff dynamics (Moses and Robinson, 2011) (moses). This implies that those studies cannot be use to describe global retreat drivers because : (i) they do not analyse the contribution of each driver. (ii) they remain too local and characterise a narrow range of forcings (e.g. climate, homogeneous lithology . . . ) "

**Par beginning line 29: it would be helpful to have some more detail on the type of study – what they measure, degree of accuracy, limitations etc. (historical maps, air photos, TLS, Lidar, photogrammetry, use of drones).**

Following this remark, we add new details: "Since Sunamura (1992)'s compilation, 26 years ago, many new quantitative studies have been published. They took advantage of several technological changes in that time interval. National mapping agencies released their aerial photography archives online, allowing to record cliff top retreat along decades. These provide contemporary surveys with a historical context. Airborne and terrestrial lidar as well as structure-from-motion (sfm) have revolutionized ad hoc surveys in geosciences, making precise geometric information available where and when required. Those methods allows to record rockfalls from cliff 35 face and assess their volumes. Software developments afforded massive 3D processing capabilities, even to non-specialists. So quantitative site studies are now addressing cliff face erosion style at centimetre-scale (e.g., Dewez et al., 2013; Earlie et al., 2015; Gulayev and Buckeridge, 2004; Letortu et al., 2015; Rosser et al., 2007; Young and Ashford, 2006). This high spatial accuracy is nowadays added to high time resolution up to 20 minutes with detection of decimetric fragments from cliff face Williams et al. (2018). Cliff recession phenomena have never been so well defined in space and time. It is now time to sort 5 through possible processes generating cliff responses."

**Page 3**

**Line 1: 'high time resolution of up to 20 minutes' – it would be helpful to say what this high temporal resolution data records – removal of individual small rock fragments from the cliff face?**

Done, see precedent paragraph.

**Line 5: 'study their relative efficiency'- not clear how this relates to linking erosion rates and external forcings – perhaps amend sentence to improve clarity.**

The amendment was done, the sentence was rephrased to: "This database is used in a new approach to link erosion rate and external forcings. It allows also to look for a relative efficiency of forcings between each other to explain erosion rates variations at global scale. "

**Line 8: 'reduces information to the largest common denominator' – yes, this may be a limitation but it is also an opportunity! It would be helpful if the paper can set out, on the basis of this study, a clear statement of the scale/resolution of study and also the important factors to record for future studies of rock coast erosion – in order to improve the resolution of the GlobR2C2 in the future. We added a new section (4.2.5) in the paper indicating which efforts must be made for future studies. Line 18-19: it woud be helpful to say here what databases are used.**

Done, explained in section therein.

**Line 27-30: sentence beginning 'It helps and the next sentence could be made clearer. For example, I am wondering if the conceptual exercise really minimises data capture? Should it be 'maximise data capture and minimise data redundancy'?**

We made particular efforts in improving this paragraph. It becomes: "Merise provides a formal methodology to describe entity-relationship data models. Each entity corresponds to a group of data framed into a table and containing different fields. The different entities are related with each other by well-defined relations. As an example the cliff entity contains information about cliff settings. Each cliff description corresponds to a line in the cliff table and contains a unique primary key to identify this line/record. The measure entity contains information about cliff erosion. Cliff and measure are related through cliff erosion. The relation between an erosion record and its corresponding cliff is made by typing the cliff primary key. This conceptual exercise allows to minimize data typing and data redundancy, to flag possible information replicates and limits ill-conceived relationships. The database structure was implemented in OpenOffice Base that can be processed in R via SQL queries. Only the geographic fields (cliff location) were digitized in GoogleEarth and exported into shapefile with a key code or primary key linked to the relational database (in the sense of data science analysis)."

**Page 4**

**Line 8: three types of sources? Are the data from scientific papers really raw data? Not clear what is meant by gridded data and tidy covariates.**

The raw data, in the sense of database design, corresponds to the information encoded into the database with as little modification as possible from the original source. The publication itself is not a raw data, but in our database we took the information from it as raw data because they are not modified from the source. The second "raw data" type are the data extracted from global reanalysis grids: the grid values are recorded in GlobR2C2 without modification. Finally, we computed physical values from other fields values. Those newly computed quantities are encoded as new fields. Data scientists call them "tidy covariates".

**Par beginning line 9: this could usefully be expanded to aid explanation. For example, is the method of measuring cliff erosion recorded and the time period over which it is measured? Figures will need to be re-numbered in order to ensure that they are referred to in the correct order.**

A more detailed explanation of what is captured in each field is given below.

**Section 2.3.2**

**Cliff lithology and description: it would be interesting to know how you have dealt with composite cliffs in the database – for example, a composite cliff may contain materials of different hardness/resistance at the toe and so marine forcing may be of reduced importance in such cases.**

Information about composite cliff was implemented in the database. The lithology entity contains a lithology name and a field called "lithology location". This field was filled with the information "toe", "head" or "everywhere". Composite cliffs represents 15% of records. In turn, only one erosion rate was associated with each cliff.

**Page 5**

**Line 2 – 4: meaning unclear and it would be helpful to recast these two sentences to improve clarity.**

These sentences are improved as: "They were characterised with a Boolean value (True/False) to be integrated in the database. True refers to the presence of fracturing/weathering mentioned in the paper. False means either that authors describe fracturing/weathering as non existent/negligible or is not mentioned in the paper."

**Line 7: not clear what is meant by 'a primary key'.**

See database design.

**Line 10: etc. – please specify what is included in the etc.!**

We specified : "The measure entity contains the erosion rate values and measurement methodology (how erosion was measured, for how long, with what threshold)."

**Line 14: suggest amending to 'estimates of volume loss to precise measurements using, for example, lidar**

The amendment was done.

**Line 15: suggest amending to '(iii) spatial extent along the coast**

The amendment was done.

**Line 23: not clear what is meant by 'the oldest method is rockfall inventory'**

We reworded in 'Initially, 3D assessment were performed based on observable, large, rockfall scars or debris apron (e.g. ...'

**Line 29: suggest amending to 'but with two caveats'**

The amendment was done.

**Line 31: it would be helpful to say how data were 'specifically treated' beforehand in order to prevent bias.**

As it is the topic of an entire section in discussion we now make reference to it.

**Page 6**

**Line 3: is it the case that faster eroding cliffs are more often sampled are more densely populated cliffs not also more often sampled by regional/national authorities?**

We don't write that faster eroding cliffs are more densely populated but that authority fund more often densely populated and/or fast eroding cliff sections.

**Line 7: suggest amending to that quality of photographs limits**

The amendment was done.

**Line 11: not clear what is meant by and produce wetting drying cycles does this mean, influences the vertical extent of wetting drying cycles on the cliff face? How about any potential influence on groundwater levels in more porous rocks?**

We rephrased in order to be clearer: "The tidal range describes the variation in height of the water surface. A consequence is that the cliff and platform undergo cyclic wetting

and drying that weakens and erodes the constituting rocks (Kanyaya and Trenhaile, 2005)."

**Line 13-14: it would be helpful to add some explanation to the harmonics.**

The amendment was done.

**Line 29: time steps**

The amendment was done.

**Line 30: spelling – below.**

The amendment was done.

**Page 7**

**Line 17: thus, 3D measures (rather than this?)**

Changed to 'the 3D measures'

**Page 8**

**Line 12: Fig. 9 is referred to but the last Fig referred to was Fig 5 – Figs 6, 7 and 8?**

In order to keep the figures at the best position in the paper we removed reference to Fig.9 here.

**Section 2.4.4:**

**It would be helpful to have some more contextual detail on the Hoek and Brown rock resistance classification that is used in the study.**

This is done. We produce a new table (Table 11) and changed the text as: "Hoek and Brown (1997) describe field estimates of rock strength and experimental uniaxial compressive strength. They describe seven grades of rock resistance, from extremely weak to extremely strong. The table describing field estimates, resistance term, compressive strength and example is given in table 11. This table is associated with our Hoek and Brown classification and associated lithologies found in the database."

**Page 9.**

**Line 6: Fig. 4 is out of synch.**

**Line 8: suggest amend to 1990s for every type of method'**

The amendment was done.

**Line 15: 6.4 km**

The amendment was done.

**Line 26: provide the number of observations for each class rather than just one.**

The amendment was done. "Hard rock (341 observations) erodes at a median rate of 2.9 4 cm.yr-1 with a Median Absolute Deviation (MAD) of 3.4 cm.yr-1. Medium resistance rock coasts (63 observations) erode at around a median value of 10 4 cm.yr-1, with a MAD of 7.8 4 cm.yr-1. Due to the small number of observation of 10 medium resistance rocks, this resistance class should be considered carefully. Finally weak rocks (403 observations) erode at with a median value of 23 4 cm.yr-1and reach rates higher than 10 4 cm.yr-1with a MAD of 25 4 cm.yr-1."

**Page 10**

**Line 17: 'amount of rainfall.'**

The amendment was done.

**Line 27: 'design allows an assessment of the drivers of erosion'?**

The amendment was done.

**Page 11**

**Section 4.2.1 See also Michoud et al. (2012) who estimated cliff retreat of the**

**"Dieppe landslide": 'activated on 17–18 December 2012 we measure a cliff re-treat up to 40 m along two active scarps over 70 m wide' (p. 415).**

Indeed, it is another good example. We however did not include it because our first example is sufficient and well documented for our purpose.

**Page 12**

**Line 10: 'this finding reflects' (remove is)**

The amendment was done.

**Line 29: amend TABLE**

The amendment was done.

**Page 13 It would be helpful to have some discussion of the importance of weathering that can be drawn on for the conclusion. It would also be helpful to make some recommendations for future studies of rock coast erosion that would help to address the data gaps identified in the compilation of GlobR2C2.**

Such a paragraph has been included in the current version of the paper (section 4.2.5).

**Figures**

**Figure 1: suggest amend to is similar to that**

The amendment was done.

**Figure 2: diagram a could usefully show the shore platform; there is no mention of faulting in the cliff settings – if it is included then it would be helpful to mention it; not clear what is meant by 'aquiferous' in the continental forcing. Diagram b seems to use only half of the 58 studies that are used in the database (there are 23 dots on the graph). Also, it is not clear what is meant by the 'authors point of view'. It would be helpful to have some more explanation either in the caption or in the text.**

The graph and its caption is modified to include shore platform and your remarks. The diagram b doesn't show all the studies used in the database but only the ones whose authors interpret and point out the erosion causes in the abstract.

**Figure 6: Hoek and Brown**

The amendment was done.

**Figure 8: typo after temperature**

The amendment was done.

**Figure 9: Woodroffe**

The amendment was done

**Bibliography:**

Gleeson, T., Moosdorf, N., Hartmann, J., van Beek, L.P.H., 2014. A glimpse beneath earth's surface: GLobal HYdrogeology MaPS (GLHYMPS) of permeability and porosity. Geophys. Res. Lett. 41, 3891–3898. https://doi.org/10.1002/2014GL059856

---

## Author Comment (AC2) · 11 May 2018

We are grateful to Larissa Naylor (reviewer 1) for her very positive and insightful comments that helped improve the paper. We agree with her view that the paper was too focused on rocky coast and we add more context and links with shore platform evolution and geomorphic context.

**General Comments**

**This paper presents a much improved, extended and comprehensive database of global coastal cliff erosion which brings together the rapidly expanding number of papers in this field, in a rigorous and comprehensive analysis. Of particular import is that their analysis allows for improved understanding of the impor-**

[Figure]

tance of rock resistance over lithology or climatic parameters as the key factor controlling erosion rates. They found that rock mass properties like joints and fractures are a fundamental control of coastal rock cliff erosion rates. They have come to these conclusions through creating a thorough, rigorous and repeatable database that can be extended through time as more papers are published. It is thus of great scientific importance and serves as a valuable tool which can be built upon. The authors are to be highly commended for their efforts. I really enjoyed reading this manuscript and once a moderate level of corrections are made, this will make a superb and much needed contribution to the literature. The key areas for improving this manuscript lie in six areas:

**a) Improving the explanation of your methods and statistical analyses. Why did you use the tests you did? Which aspects did you test statistically?**

For the purpose of this paper, we chose to follow a classical exploratory data analysis (EDA) scheme. An EDA is a prerequisite before embarking on more sophisticated methods such as machine learning algorithms. Machine learning classification and prediction is on our agenda and has been partly but we keep these results for a future paper. In the present paper, we tested the relation between a dependant variable (erosion rate) and (supposed) independent variables (external forcings). The relations between forcings was explored through principal component analysis. They appeared to be often strongly correlated. Concerning relation between erosion and continuous values, the non-normal distributions make us choose a non-parametrical approach. Spearman's rank correlation was chosen to evaluate the monotony of the relation.

**b) Considering 'dating techniques' as an additional category of how erosion is measured and adding this category into your analysis (eg Figure 5) or explaining where this content best fits in your classification from 1D to 3D studies.**

Erosion rates measured with Âń dating techniques Âż were encoded into the database (e.g. Choi et al., 2012; Hurst et al., 2017; Regard et al., 2012) . However we choose

not to include them in the analysis because they don't represents the same processes and are dependent of eustatic variations. Those studies are generally transect type and would be classified as "1D" techniques.

**c) Threshold, non-linear behaviour of coastal rock cliff erosion. Many of the types of cliffs included often display threshold-driven, non-linear behaviour. Whilst I appreciate you needed to standardise your reporting of erosion to mm/yr-1, I also wonder if it is possible to evaluate the degree of stochasticity /non-linearity in the database. For example, it may be that certain rock resistance types are more prone to non-linear, stochastic erosion events or that the temporal frequency between erosion events varies by rock resistance category or another parameter. Finding a clever way including this alongside your mm/yr-1 would improve awareness of the behaviour of these systems for risk managers, hazard scientists and geomorphologists alike.**

The question of stochasticity/ non-linearity of erosion is an interesting one. However, it is difficult to approach it with our database. In fact, the major part of our data corresponds to averaged erosion rates over decades, mainly computed through comparison of aerial photographs. Evaluating the non linearity would need a more regular temporal monitoring of cliffs. This kind of approach is only really possible since lidar and SfM methods have become available to examine cliff face erosion. These techniques go back 10 years ago, which is a very short period of time to convince oneself that non-linear behaviours were reliably observed. Then, threshold detection requires frequent surveys for a long duration in order to build rockfall scar inventories. This is a lot of scientific effort to sustain acquisition, consistent processing and funding over time. Only very few scientific teams have been able to do this so far. The most prominent one is probably Nick Rosser's group in Durham, UK with whom we have started to collaborate.

One way to evaluate the non linearity is to explore magnitude/frequency laws. Those laws appears to be power law and have been explored for rockfalls, specially for

continental cliffs (Barlow et al., 2012; Brunetti et al., 2009; Dussauge et al., 2003; Dussauge-Peisser et al., 2002). One way to evaluate the different stochasticity of rockfall could be to compare the coefficients of those power laws. However, the coefficients appear to be highly dependant on the observation duration (Dussauge-Peisser et al., 2002), spatial and temporal resolution (Williams et al., 2018).

In short, it is a very insightful question to which we cannot respond in a satisfactory fashion. We nevertheless aknowledge the question of rockfall stochasticity in section 2.4.1 Integration of punctual records, and further discuss it in 4.2.1 Erosion rates, study duration and stochastic behaviour.

**d) Wider context. In places, the analysis and discussion of this paper is too narrowly focussed on coastal rock cliff erosion, rather than drawing on evidence from recent shore platform research which displays similar trends around the importance of geological contingency, the importance of rock mass properties and weathering/rock breakdown (bio/chem/phys) processes helping prepare rock coast landforms for erosion. This includes the early conceptual models of cliff erosion by Sunamura as well as recent papers on rocky shore platforms.**

We added a last paragraph in discussion 4.2.4 Cliff retreat vs platform evolution and rock coast erosion. We also improved fig. 2 to show the platform more explicitly.

**e) Figure 1 and your discussion of it shows the importance of the wider geomorphic context in influencing erosion rates. This does not appear to be taken into consideration in the current version of your model. It would be useful for the authors to explore how this may be possible, so that a global analysis of how submarine to cliff-top coastal landforms vary around the globe and how this affects erosion rates. For example, what proportion of cliffs globally are currently shielded by offshore features such as those in part of Figure 1? Does this vary by rock resistance of the cliffs or are other factors influencing this? I realise that much of this may be beyond the scope of your current paper, but it may be use-**

ful to signpost this in your current paper, perhaps using data from both parts of
Figure 1 as an example to illustrate how cliff erosion rates are modified by their
wider geomorphic context, and thus are partly geomorphologically controlled.

We agree that taking wider geomorphologic context would add a lot to the model.
Currently, with the notable exception of cliff height, we don't take geomorphology into
account neither for the cliff nor submarine geomorphology. It would be a big challenge
to take it into account for two main reasons. The first one is to find appropriate geo-
morphologic descriptors that make consensus across the community, or propose our
own, which could be regarded as a curiosity. The second challenge is to find available
auxiliary data to get those descriptors consistently for all sites.

**f) Lastly, it would be useful to signpost the wider significance of your work for
coastal hazards scientists, geologists and in the context of changing storminess
and sea level rise. It also would be helpful to highlight the potential to extend the
database to include shore platform erosion rates. This would help show the
wider relevance and import of your work.** Ok, this was done in section 4.2.5 Toward
a new rocky coast cliff research agenda

**Specific Comments (SC), Technical Comments (TC)**

**Title**

**SC - You may wish to change the title to better capture the global database /anal-
ysis that is, to me, a significant strength of your paper and a very strong addition
to the literature.**

Thank you for this advice. The title was changed to: "GlobR2C2 (Global Recession
Rates of Coastal Cliffs): a global relational database to investigate coastal rocky cliff
erosion rates variations."

**Abstract**

**SC - Show the wider relevance of your important work here**

We added a final sentence to allude to several impact of this research:

In this first version, GlobR2C2, with its current encompassing vision, has broad implications. Critical knowledge gaps have come to light and prompt a new coastal rocky shore research agenda if one day we hope to answer such questions as coastal rocky shore response to sea-level rise or to increased storminess.

**Introduction:**

**TC - First sentence needs reference and second sentence needs a direct quotation.**

We added reference to Moses and Robinson (2011) for the first sentence.

**SP – wave-cut vs shore platform needs a little more discussion**

We think the shore platform is already present in our description of the processes leading to cliff erosion. To make it clearer to both reviewer we improved the figure 2 and add some sentences within the introduction to better articulate between the shore platform and rock coast erosion.

**TC – para 25 Fig 3 or Fig 5?**

The figure reference was removed here.

**TC – para 30 cite Viles 2017 Geomorphology**

This reference was added

**Method:**

**SC – define your boundary conditions and cite Kennedy who first used this term explicitly in rock coast geomorphology**

Changed to

"However, marine and continental forcings conditions are often reported in a very het-

erogeneous fashion."

**SC – systematic search method needs improving, this can either be quite simple as per Figure 1 in Naylor et al. 2010 or following the more detailed PRISMA method (Moher et al. 2015) stemming from medical science.**

We did not understand this comment

**TC – Merise needs a year, pg. 3 para 25**

The reference is Tardieu et al. (1985)

**SP – pg 4, Para 10 sentence 1 examples adding would be helpful to aid understanding of your database.**

In order to improve clarity, an example was added as the first paragraph of section 2.2 Database design. "As an example the cliff entity contains information about cliff settings. Each cliff description corresponds to a line in the cliff table and contains a unique primary key to identify this line/record. The measure entity contains information about cliff erosion. Cliff and measure are related through cliff erosion."

**TC – pg, 4 para 5, first sentence could be reworked**

The wording was deliberately casual but we rephrased it as required in section 2.3.2 Cliff and lithology description

Cliff geology may exhibit a possibly very complex set of lithologic types, contact relationships, inherited tectonic structures and overprinted weathering and authors. . .

**SP – section 2.3.1. a) Only English is mentioned here but Spanish and French is mentioned earlier. B) define your search method and strings (perhaps as supplementary material), this will make this part of your work reproducible and improve rigour.**

a. Peer reviewed articles are in English and Âń white literature Âż is in English, French

or Spanish b. Our search method was :

- We started from a corpus of articles identitified by an early undergrad student's work

- The references cited in this initial set of articles we then explored

- Searches where then launched in bibcnrs (national French research center bibliography engine) with keywords "erosion", "sea", "cliff", "rocky"

- Finally, this was completed with an email call to the coastal community via "coastal list"

This procedure may not be as rigorous and reproducible as Naylor would have wished for. But even if it is an organic growth of knowledge, the corpus of data is now contained and structured in GlobR2C2 and any new reference can be checked against existing records.

**SP – 2.3.4 add Hurst et al. 2017 as reference for 1000s of years scale**

The citation was added.

**SP - 2.3.5 last sentence is unfinished**

Sentence was completed: "We discuss this choice in discussion section."

**SP – 2.3.9 can you validate your assertion in the last sentence?**

The first attempt at global scale has been verified to be satisfactory (sentence before), but we cannot estimate the accuracy of cliff height indicated in the publications (maybe on the order of 10 meters).

**SP – 2.4.4 Not all of your core readers will be familiar with the Hoek Brown criterion as it is a geotechnical/engineering criteria. I recommend you add some**

**background information and some rationale for why this was the best metric to use. Here it would be good to explain why Selby 1980 is less suitable than Hoek Brown.**

This is done. We produce a new table (Table 1) and changed the text as: "Hoek and Brown (1997) describe field estimates of rock strength and experimental uniaxial compressive strength. They describe seven grades of rock resistance, from extremely weak to extremely strong. The table describing field estimates, resistance term, compressive strength and example is given in table 1. This table is associated with our Hoek and Brown classification and associated lithologies found in the database."

**Section 3:**

**SP – 3.3 See comment above about dating methods.**

**SP – 3.4.1 fewer medium resistance rock studies, perhaps make this as a suggestion for future research in your conclusions, along with the present geographic limitations?**

The few records concerning medium resistance rocks is due to several reasons: 1. Medium resistance rocks concern a smaller spectrum of rock types than weak and strong ones (see table 1 for unique lithological names). Weak resistance rock varies from extremely weak (can be peeled with a nail) to weak. Our weak and strong rocks actually aggregate 3 class of Hoek and Brown criterion (extremely/very/ weak, and equivalent for strong) (see table 1). 2. The large majority of erosion rates in hard rock cliff is brought by the systematic survey of CEREMA along the French coastline (265 values over 343, 77%). Despite these justifications, we added a mention of this objective I section 4.2.5 Toward a new rocky coast cliff research agenda

**Section 4:**

**SC- Para 5, page 11 - more detail on this conference, a specific pers comm would help here too.**

We cannot locate to what this comment refers to .

**SC- Weathering, jointing, discontinuities – Sunamura included these parameters in his early conceptual models of rock cliff, rock coast and shore platform erosion, showing how they contributed to the reducing the resisting force of rocks. The influence of these on erosion processes and rates has been more recently discussed for rocky shore platforms (See Cruslock et al. 2010, Naylor and Stephenson, 2011, Stephenson and Naylor 2012) and biology (Naylor et al. 2012).**

The citations are added.

**SC - 4.2.1 para 20, this is where the threshold, non-linearity comment above relates.**

**SC - 4.2.3 pg 12, para 10, I recommend you refer to Kennedy et al. 2014 here as this volume has no chapter on Africa, which accords with your analysis of rocky cliffs. Doing so would strengthen this point.**

The reference to Kennedy et al. 2014 was added.

**SP – pg 12, para 20, does this mean it relates only to softer rocks? Please clarify.**

The paragraph was modified to be clearer: "Studies also focus on fast eroding coasts because they represent bigger risks and also because of methodological limitation. Indeed, the French CEREMA study brings the majority of erosion values for hard rocks (265 values over 343, 77%) and medium rocks (47 values over 66, 71%). Without this systematic study soft rock represents 75% of measured cliff retreat. This fact biased the analysis by mostly documenting erosion distribution in higher values. The weight of this bias can be approached thanks to the French CEREMA study."

**TC - Pg 12, para 25, I think this is table 2?**

The reference was table 2, the text was modified.

**SC - Page 13, para 25 there are many newer rock coast evolution models including consideration of the impacts of climate change (e.g. Limber, Ashton, Trenhaile) that are worth looking at to improve your link to modelling.**

We just indicated some examples here. We added a citation to Limber et al. (2014).

**Technical comments for the whole the Manuscript: Minor improvements to your English is needed occasionally throughout the manuscript The manuscript has be minutely checked for English. Measure often needs to be measurement**

The text was checked to correct this.

**Page 13 – inshore could be confused with 'inshore waves'; I recommend using terrestrial instead.**

This amendment was done

**Page 20, what does Q83 refer to?** To the 83% quantile, modified. Also add a final sentence, or extension to it that shows which rock categories this relates to.

**In a few places you talk about rocky coast erosion, your topic is coastal rocky cliff erosion. For clarity about your scope and the contents of your paper, the latter term should be used throughout.** Your remark was taken into account and the term coastal rocky cliff was used.

Bibliography: Barlow, J., Lim, M., Rosser, N., Petley, D., Brain, M., Norman, E., Geer, M., 2012. Modeling cliff erosion using negative power law scaling of rockfalls. Geomorphology 139–140, 416–424. https://doi.org/10.1016/j.geomorph.2011.11.006 Brunetti, M.T., Guzzetti, F., Rossi, M., 2009. Probability distributions of landslide volumes. Nonlinear Process. Geophys. 16, 179–188. Choi, K.H., Seong, Y.B., Jung, P.M., Lee, S.Y., 2012. Using cosmogenic 10Be dating to unravel the antiquity of a rocky shore platform on the west coast of Korea. J. Coast. Res. 28, 641–657. Dussauge, C., Grasso, J.-R., Helmstetter, A., 2003. Statistical analysis of rockfall volume distributions: Implications for rockfall dynamics. J. Geophys. Res. Solid Earth 108.

https://doi.org/10.1029/2001JB000650 Dussauge-Peisser, C., Helmstetter, A., Grasso, J.-R., Hantz, D., Desvarreux, P., Jeannin, M., Giraud, A., 2002. Probabilistic approach to rock fall hazard assessment: potential of historical data analysis. Nat. Hazards Earth Syst. Sci. 2, 15–26. Hurst, M.D., Rood, D.H., Ellis, M.A., 2017. Controls on the distribution of cosmogenic 10Be across shore platforms. Earth Surf. Dyn. 5, 67–84. https://doi.org/10.5194/esurf-5-67-2017 Regard, V., Dewez, T.J.B., Bourlès, D.L., Anderson, R.S., Duperret, A., Costa, S., Leanni, L., Lasseur, E., Pedoja, K., Maillet, G.M., 2012. Late Holocene seacliff retreat recorded by 10Be profiles across a coastal platform: Theory and example from the English Channel. Quat. Geochronol. 11, 87–97. https://doi.org/10.1016/j.quageo.2012.02.027 Williams, J.G., Rosser, N.J., Hardy, R.J., Brain, M.J., Afana, A.A., 2018. Optimising 4-D surface change detection: an approach for capturing rockfall magnitude–frequency. Earth Surf. Dyn. 6, 101–119. https://doi.org/10.5194/esurf-6-101-2018

---

## Author Comment (AC3) · 14 May 2018

Dear associate editor,

Please find enclosed a reviewed version of our manuscript with a new title (as suggested by reviewers) called "GlobR2C2 (Global Recession Rates of Coastal Cliffs): a global relational database to investigate coastal rocky cliff erosion rates variations". The article was modified thanks to the very insightful comments of Larissa Naylor and Cherith Moses (two detailed answers, one for each reviewer are already posted). Both reviewers highlighted the scientific significance of this article for the database it describes and the necessary research agenda the analysis revealed. The reviewers suggested a moderate level of correction. The context was widened, we included more

details about coupled shore platform and cliff evolution and linked it to coastal rock cliff. We also highlighted the limitations of existing studies and knowledge gaps concerning in existing literature to understand the evolution of rocky coastal cliffs. As suggested, the English was improved as best we could and errors corrected. We provide two improved versions of the article, one with changes in colour and a final version. We add a supplementary material consisting in an extract of GlobR2C2 database containing the data discussed in the paper, in excel format.